# Efficacy and safety of Cheonwangbosim-dan (Tian Wang Bu Xin Dan) for treatment of mild cognitive impairment: A randomized placebo-controlled pilot trial

Jae-Hong Kim[1,2]*, Ae-Ran Kim[3], Bok-Nam Seo[3], Gwang-Cheon Park[2], Jeong-Soon Lee[4]

**1** Department of Acupuncture and Moxibustion Medicine, College of Korean Medicine, Dongshin University, Naju, Republic of Korea, **2** Clinical Research Center, Dongshin University Gwangju Korean Medicine Hospital, Gwangju, Republic of Korea, **3** Clinical Research Coordinating Team, Korea Institute of Oriental Medicine, Daejeon, Republic of Korea, **4** Department of Nursing, Christian College of Nursing, Gwangju, Republic of Korea

* nahonga@hanmail.net

## Abstract

Mild cognitive impairment (MCI), an intermediate condition between healthy cognitive changes due to aging and dementia, is a risk factor for dementia. Cheonwangbosim-dan (CWBSD) is a traditional herbal medicine widely used to improve mental and physical illness in East Asia. We aimed to investigate the safety and efficacy of CWBSD in the treatment of MCI. This clinical trial adopted a double-blind, single-center, parallel-arm, prospective, randomized controlled pilot study design with a full analysis set. Forty-eight participants with MCI were randomized to the control or CWBSD groups. The participants were administered either placebo or CWBSD once daily (at a dose of 3 g) for twenty-four weeks. The primary outcome measure was a change in the Korean version of the Montreal Cognitive Assessment Scale (MoCA-K). The secondary outcome measures included memory, quality of life, depression, and level of activity in daily living. Safety measures were conducted by comparing the occurrence of adverse events and changes in blood chemistry parameters, pulse rate, and blood pressure between the two groups. After 24 weeks of treatment, no significant differences in the changes in the total MoCA-K scores were detected between the two groups. Changes in secondary outcomes were not significantly different between the two groups. In the safety evaluation, there was no significant difference between the two groups with regard to the occurrence of adverse events and changes in blood chemistry parameters, blood pressure, and pulse rate. We demonstrated that CWBSD treatment may be safe and does not significantly improve cognitive function, quality of life, daily activities, or depression in individuals with MCI. Further studies are necessary to validate the efficacy and safety of CWBSD in the treatment of MCI.

**Data availability statement:** All relevant data are within the manuscript and its Supporting Information files.

**Funding:** This work was supported by the National Research Foundation of Korea (NRF) grant funded by the Korean government (MSIT) (No. NRF-2021R1A2C2007041). The sponsor has no role in the study design and will have no role in data collection, analysis, or interpretation; writing of the report; or the decision to submit the report for publication.

**Competing interests:** The authors have declared that no competing interests exist.

## Trial registration

Clinical Research Information Service KCT0006787.

## Introduction

Mild cognitive impairment (MCI) is an intermediate stage between normal cognitive aging and dementia. It is characterized by a decline in one or more cognitive domains, including memory, problem-solving, attention, concentration, language, executive functions, or visuospatial abilities, while daily activities remain largely intact [1,2]. It is a significant risk factor for dementia [3,4]. Older adults (≥ 65 years) with MCI have an elevated risk of developing dementia within three years [5]. Many individuals with MCI may revert to normal cognition or remain stable, whereas others may exhibit progressive decline, culminating in dementia [6]. Therefore, management and early detection of MCI could aid in the prevention of dementia [7,8].

MCI management aims to hinder, prevent, or even reverse the progression of dementia. Recommendations for management can be classified into four categories: counseling, pharmacological therapies, interventions for risk reduction, and non-pharmacological therapies [9]. No pharmacological interventions for MCI supported by high-quality evidence are currently recommended [10]. However, managing modifiable risk factors or non-pharmacological interventions, such as regular exercise and cognitive training, may be effective in managing MCI [7,9,11].

Historically, herbal medicines, including single herbs and preparations, have been used to treat cognitive impairment [12,13]. Recent randomized controlled trials (RCTs) [14–16] and systematic reviews [17,18] have suggested the potential of herbal medicines for the treatment of MCI. Cheonwangbosim-dan (CWBSD), also known as Tennohosintan in Japan and Tian Wang Bu Xin Dan in China is a traditional herbal medicine composed of 15 medicinal herbs. It has been widely used in Korea to treat conditions such as insomnia, depressive and anxiety disorders, and heart palpitations. Given its potential effects on neurological health, its possible role in cognitive function has been investigated [19–21]. Preclinical animal studies have reported that CWBSD exhibits neuroprotective effects [22] and exerts therapeutic effects in AD [23]. CWBSD has been widely used to treat psychological and behavioral symptoms associated with AD [24] and MCI in Korea [25].

Although CWBSD can be used to treat MCI, clinical evidence demonstrating its potential is insufficient. Therefore, we conducted a randomized placebo-controlled pilot trial to evaluate the safety and efficacy of CWBSD in the treatment of MCI.

## Materials and methods

This trial adhered to the Korean Good Clinical Practice guidelines and principles of the Declaration of Helsinki. The detailed methods have been previously reported [26].

### Ethical considerations

The trial protocol (version 1.2) was approved by the Institutional Review Board (IRB) of Dongshin University Gwangju Korean Medicine Hospital

(Number: DSUOH-2021–004; date: November 19, 2021). This study was registered with the Clinical Research Information Service (cris.nih.go.kr; date: November 26, 2021; number: KCT0006787). The potential risks and purpose of the trial were explained to all participants who provided written informed consent before participation.

## Study design

This trial was a pilot, prospective, parallel-arm, double-blind, single-center RCT. Forty-eight enrolled participants were randomly allocated to the control group (placebo; n = 24) or the CWBSD group (CWBSD; n = 24). Participants were educated on exercise and self-management and administered either CWBSD or a placebo once daily for 24 weeks.

Efficacy outcome measurements and blood chemistry tests were conducted at baseline (Week 0), 12 weeks after baseline (Week 12), and 24 weeks after baseline (Week 24; intervention completion).

## Participant recruitment

Participants were recruited at Dongshin University Gwangju Korean Medicine Hospital in the Republic of Korea from May 13, 2022, to January 20, 2023. Our study ended on July 20, 2023. Candidates were screened using various inclusion and exclusion criteria.

## Participation

**Inclusion criteria.** The inclusion criteria were as follows: 1) older adults, aged between 55 and 85 years; 2) experienced impaired memory persisting for at least three months, meeting the diagnostic criteria for MCI [27] i.e., a) self- or informant-reported cognitive decline, b) preserved daily functioning, c) impairment in one or more of the six cognitive domains, d) absence of dementia criteria, and e) no MCI attributed to delirium or other psychological disorders [e.g., depression]; 3) attained a score of 20–23 on the Korean version of the Mini-Mental State Examination (K-MMSE); 4) achieved a score of 0–22 on the Korean version of the Montreal Cognitive Assessment scale (MoCA-K); 5) scored between 0 and 18 on the Korean version of the Geriatric Depression Scale (GDS-KR); 6) received a Global Deterioration scale score of 2 or 3; 7) demonstrated fluency in the Korean language for reliable evaluation; and 8) provided voluntary consent.

**Exclusion criteria.** The exclusion criteria were as follows: 1) Individuals diagnosed with vascular dementia according to the NINDS-AIREN criteria or AD as per the NINCDS-ADRDA criteria; 2) those with a recent (within the past 12 months) diagnosis of brain disease, confirmed via brain magnetic resonance imaging or computed tomography; 3) participants with a history of structural brain lesions that could cause cognitive impairment, such as stroke, traumatic brain injury, intracranial space-occupying lesions, or congenital mental retardation; 4) individuals receiving treatment for MCI, including Korean Medicine (KM), drug, or cognitive training, within 4 weeks before screening; 5) a history of treatment for drug or alcohol dependency or mental illness, including depression, severe anxiety, or schizophrenia, within six months prior to screening; 6) those with serious medical conditions like Huntington disease, Parkinson's disease, cancer, central nervous system diseases, liver diseases, cardiovascular diseases, kidney diseases, or multiple sclerosis; 7) uncontrolled diabetes (fasting blood sugar ≥ 180 mg/dL) or hypertension (blood pressure ≥ 170/100 mmHg); 8) renal dysfunction, defined as creatinine levels at least twice the normal level; 9) similarly, hepatic dysfunction, indicated by alanine aminotransferase or aspartate aminotransferase levels at least twice the normal level; 10) difficulties in undergoing outcome measurement due to visual and hearing impairments; 11) a history of hypersensitivity to the compositions of the placebo or CWBSD; 12) individuals with a history of nausea, vomiting, anorexia, and gastrointestinal diseases that could affect the absorption of the placebo or CWBSD; 13) participation in another clinical study within 4 weeks prior to screening or concurrent participation in another study; 14) pregnancy or breastfeeding.

**Dropout criteria.** Participants were discontinued from the trial under the following conditions: 1) incidence of any serious adverse event (SAE) (i.e., death or danger to life has occurred, requiring hospitalization, prolonged

hospitalization, or permanent or significant disability and impairment); 2) incomplete data critical for impact assessment; 3) revocation of consent for participation; and 4) termination of participation determined by the principal investigator or IRB.

### Randomization and blinding

Following an initial screening, the enrolled participants received serial codes generated using SPSS software version 21.0 and were randomly allocated to either the control or CWBSD group. Randomization numbers were sealed in opaque envelopes and stored in a double-locked cabinet. An independent investigator, who was not involved in the trial, was responsible for generating the allocation sequence, whereas a clinical research coordinator managed participant enrollment and group assignments.

Throughout the trial, all participants and outcome assessors who received training in the assessment tools employed in our study evaluated the efficacy and safety outcomes, and the investigators remained blinded to the group assignments. The pharmaceutical company tasked with producing the placebo and CWBSD ensured that both were indistinguishable in taste, smell, packaging, and appearance and labeled them according to the randomization numbers. Unblinding was permitted under IRB approval, specifically in the event of an SAE.

### Intervention

The CWBSD pellet, a standardized extract, is a brown powdered mixture of medicinal herbs formulated into a solid pellet form. CWBSD (Product Name: Soon Shim Hwan) is an over-the-counter (OTC) drug packaged in 3 g per dose and administered orally as a single daily administration. The participants were instructed to swallow the pellets with water. The detailed components, appearance, and packaging are presented in the S1-S5 Files. Placebo and CWBSD pellets, which were identical in smell, taste, and appearance, were produced by Hanpoong Pharm and Food Co. Ltd. (Seoul, Republic of Korea).

CWBSD or placebo pellets were administered once daily (at a dose of 3 g) for 24 weeks. CWBSD has been widely used for insomnia [19–21]; therefore, we advised the participants to take CWBSD or placebo once daily after dinner.

Participants were required to visit the study site every six weeks to receive a 6-week supply of either CWBSD or placebo. The clinical research coordinator (CRC) conducted at least one phone check between visits to monitor medication intake, and the participants were instructed to return any unused medication during their second to fifth visits. Adherence was assessed using the pill count to ensure compliance.

In addition to the pharmacological intervention, all participants received the same education on self-management techniques and physical exercises to prevent the progression of MCI. An investigator conducted an education session with the brochure at each visit for 10–20 min. The educational program included recommendations for exercise, diet, and lifestyle management to support cognitive health. The exercise component of the program emphasized establishing regular exercise habits, such as daily mild stretching and toning exercises and walking for at least 30 min. The participants were provided instructions to perform simple exercises focusing on stretching, balance, coordination, and relaxation to help maintain cognitive function. Dietary recommendations emphasized the consumption of nutrient-rich foods beneficial for cognitive function, such as fruits, fatty fish (e.g., salmon, mackerel, and sardines), whole grains, vegetables, soy products, and legumes, while advising participants to limit the intake of sweeteners, unhealthy fats, and refined carbohydrates, which may negatively affect cognition. Lifestyle management focused on modifiable risk factors, including the management of underlying conditions such as hypertension, diabetes, and dyslipidemia; stress management; cognitive training; smoking cessation; alcohol avoidance; and maintaining social engagement with family and community members. These guidelines were delivered through individual educational sessions (10–20 min) during each visit, supplemented by a standardized brochure based on the dementia guidebook published by the Ministry of Health and Welfare in Korea. However, adherence to these recommendations was not monitored.

During the trial period, the use of other treatments potentially beneficial for MCI symptoms or drugs containing any form of CWBSD, such as memantine, rivastigmine, donepezil, or galantamine, was prohibited. Treatment for other conditions that did not influence cognitive function was permitted.

## Outcome measurements

**Efficacy outcome.** The primary efficacy outcome was the difference in the changes in the MoCA-K total scores at 24 weeks post-baseline between the groups. Secondary outcomes included intergroup differences in the MoCA-K total scores at 12 weeks post-baseline, along with variations in the Korean version of the Alzheimer's Disease Assessment Scale-cognitive subscale-3 (ADAS-K-cog 3), GDS-KR scores, European Quality of Life Five Dimension Five Level Scale (EQ-5D-5L), Korean Activities of Daily Living (K-ADL), and Korean Instrumental Activities of Daily Living (K-IADL) at 12 and 24 weeks post-baseline.

The Montreal Cognitive Assessment Scale is a commonly used instrument for screening MCI in older individuals and measuring cognitive function across various domains. It is a 30-point test that evaluates the cognitive domains of executive function, concentration, attention, working memory, short-term memory recall, visuospatial abilities, orientation to place and time, and language [28]. The MoCA-K is a brief, suitable, and reliable tool for screening individuals with MCI. The optimal cut-off score of the MoCA-K (22/23) is three points lower than that of the original MoCA, which may be due to fewer years of education and cultural differences [29].

The Geriatric Depression Scale is a widely used self-report screening instrument to assess depressive symptoms in older adults. It was specifically designed to differentiate between the symptoms of depression and dementia in this demographic group. The GDS-KR is a valid and reliable questionnaire for screening major and minor depressive disorders in older people [30].

The Korean version of the Alzheimer's Disease Assessment Scale is a reliable and valid tool for diagnosing AD and evaluating its severity [31]. The ADAS-K-cog 3, comprising orientation, word recall, and word recognition tasks from the ADAS-K-cog 11, is particularly effective for MCI, as it does not exhibit ceiling effects. This tool focuses exclusively on memory assessment [32].

The EQ-5D-5L is a widely used patient-reported health utility instrument for evaluating quality of life. The Korean version of the EQ-5D-5L is a reliable and valid tool for measuring health-related quality of life in the general population of South Korea [33]. The quality weight for EQ-5D-5L was based on previous studies [34].

K-IADL and K-ADL are valid and reliable instruments for assessing basic and complex daily activities in older people [35,36].

**Safety outcome.** Safety evaluations were conducted by monitoring the incidence of adverse events (AEs) and alterations in blood chemistry parameters, pulse rate, and blood pressure across both groups. All AEs and SAEs were actively monitored throughout the study. Participants were systematically asked about any adverse symptoms at each visit and encouraged to report any unexpected symptoms between visits via phone contact with the CRC. Reported AEs were documented in detail, including the time of occurrence, severity, potential causal relationship with the trial medication, and any treatment provided. The CRC reported all recorded AEs and SAEs to the principal investigator and IRB to ensure appropriate follow-up and action.

## Sample size estimation

We could not find any existing data showing the efficacy of CWBSD using MoCA-K, which is required for sample size calculations. Therefore, we adopted a pilot study design. The appropriate sample size for the pilot study was > 12 [37]. As our study was a pilot trial, a formal sample size estimation was not conducted. Instead, considering the sample size of a previous pilot study that explored the efficacy and safety of Chinese herbal medicine for MCI [16], 48 participants were recruited and equally distributed, with 24 participants assigned to each group. The sample size may have been insufficient to determine the effects of CWBSD on MCI. The findings of our study provide preliminary data on the effects of CWBSD on MCI.

## Statistical analyses

Statistical analysis protocol was developed by referring to previous clinical trials of herbal medicine for MCI [38,39]. Statistical analyses were revised according to IRB directives during the trial period. Efficacy evaluation was performed using a full analysis set. Missing values were imputed using the last observation carried forward (LOCF) technique. An independent statistician analyzed the final data at a 5% (two-tailed) significance level using IBM SPSS software V. 21.0 software package (IBM Corp., Armonk, NY, USA). Continuous variables are expressed as mean±standard deviation or median with interquartile range, whereas categorical variables are expressed as frequencies or percentages. An interim analysis was not performed.

A Shapiro–Wilk test was performed to check the normality of the data distribution. Efficacy outcome changes were assessed using an independent *t*-test (parametric) or Mann–Whitney U test (non-parametric) for between-group comparisons. A sub-analysis was not performed.

The frequency of AEs was compared between the groups using the chi-square test for safety outcomes. Changes in blood chemistry parameters, blood pressure, and pulse rate were evaluated using the Mann–Whitney U test or independent *t*-test for intergroup comparisons.

## Results

### Participants

Participants were recruited between May 13, 2022, and January 20, 2023. Our study ended on July 20, 2023. Approximately 284 older adults were evaluated for eligibility, and 236 individuals were excluded. Forty-eight participants were randomly included in the trial and assigned to the CWBSD group (n=24) or the control group (n=24). Six participants in the CWBSD group and four in the control group dropped out during the medication phase because of withdrawal of consent. The withdrawal of consent was not due to AEs but to a simple change of mind. Data from 48 participants were used for the final efficacy and safety analyses. (Fig 1).

### Baseline study variables and characteristics

The baseline study variables and demographic characteristics of the participants are presented in Table 3. No significant differences in the baseline study variables or demographic characteristics, except for age, were detected between the two groups (p>0.05), and there was a significant difference in the age of the participants between the two groups (p=0.01) (Table 1).

### Efficacy evaluation

We compared the degrees of change in the MoCA-K, GDS-KR, K-ADL, K-IADL, ADAS-K-cog-3, and EQ-5D-5L scores between the two groups (Week 12 vs. Week 0 and Week 24 vs. Week 0) using the independent *t*-test or Mann–Whitney *U* test. (Tables 2 and 3)

**Korean version of the Montreal Cognitive Assessment Scale (MoCA-K).** The CWBSD group did not exhibit significant improvement in comparison to the control group regarding the MoCA-K total score at Week 24 vs. Week 0 and Week 12 vs. Week 0 (Table 2 and Fig 2).

**Other secondary efficacy outcomes.** No secondary efficacy outcome significantly differed in the between-group comparison (Table 3; Fig 3).

### Safety evaluation

Safety outcomes were assessed by comparing the variances in blood chemistry test parameters, blood pressure, pulse rate, and incidence of AEs/SAEs at 24 weeks post-baseline (Week 24) between the two groups. SAE did not occur in either group. One AE was reported in both the CWBSD and control groups. Specifically, one participant in the CWBSD

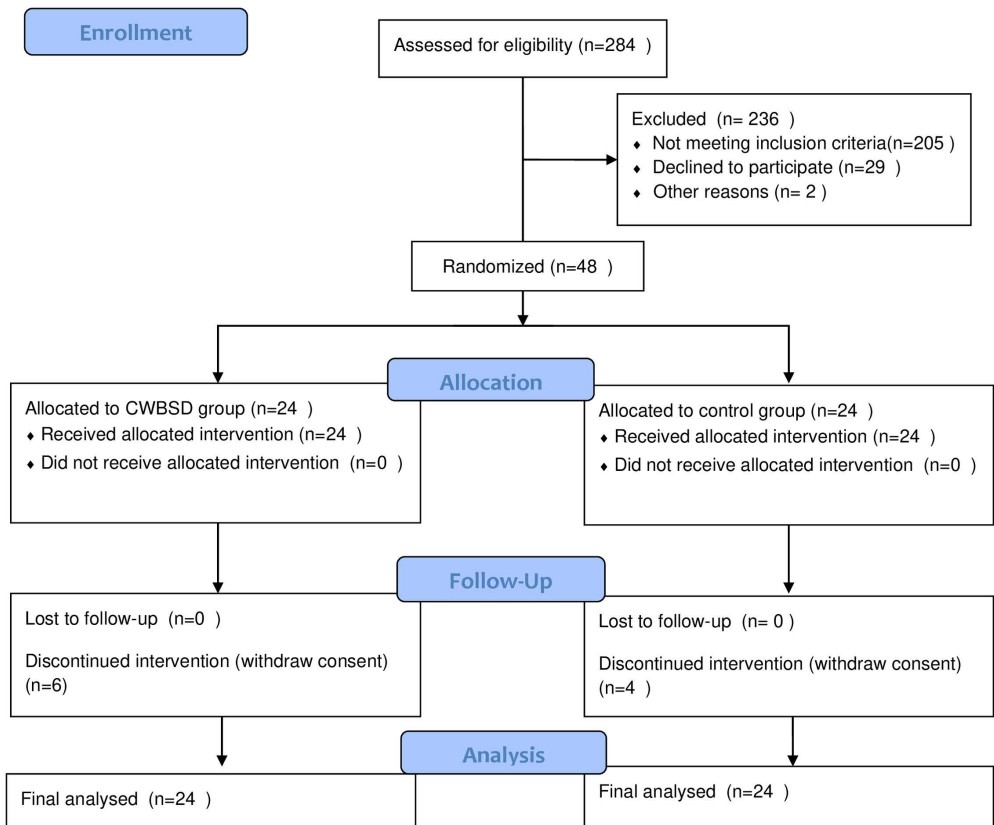

**Fig 1. CONSORT 2010 flow diagram.**

group experienced knee pain, whereas another participant in the control group reported finger pain (Table 4). Both AEs resolved spontaneously without intervention and were deemed unrelated to the trial medication. Furthermore, no significant drug-induced alterations were observed in blood pressure, pulse rate, or blood chemistry test parameters. A notable exception was potassium levels, which exhibited a significant increase in the CWBSD group (Week 0, 4.6 [4.1–4.7]; Week 24, 4.6 [4.2–4.7]; p = 0.033) compared to the control group (Week 0, 4.5 [4.2–4.7]; Week 24, 4.3 [4.1–4.6]). However, it is pertinent to note that the elevated potassium levels remained within normal ranges. Detailed blood chemistry test parameters are presented in the S6 File.

## Medication compliance

Medication compliance was calculated using the following formula:

$$\frac{\text{Actual dosage}}{\text{mandatory dosage}} \times 100 \ (\%)$$

There was no significant difference in medication compliance between the two groups at 24 weeks (Table 5).

## Discussion

The main findings of our study revealed that CWBSD had an acceptable safety profile. However, it failed to show significant improvements in cognitive function, quality of life, daily activities, and depression were not observed in individuals

**Table 1. Homogeneity tests for baseline study variables and demographic characteristics for 48 participants.**

| Dependent Variables | CWBSD (n = 24) | Control (n = 24) | *p*-value |
|---|---|---|---|
| **Demographic characteristics** | | | |
| Age (y) | 71.17 (5.62) | 67.33 (4.03) | 0.010* |
| Gender (Female) | 21 (87.50%) | 24 (100%) | 0.188‡ |
| Education (years) | 9.71 (3.94) | 9.67 (2.60) | 0.966* |
| smoking | 0 (0%) | 0 (0%) | – |
| **co-morbidities (co-medication)** | | | |
| Hypertension | 9 (37.50%) | 8 (33.33%) | 0.763‡ |
| Diabetes mellitus | 3 (12.50%) | 3 (12.50%) | 1.00‡ |
| Dyslipidemia | 10 (41.67%) | 6 (25.00%) | 0.221‡ |
| Heart disease | 1 (4.17%) | 3 (12.50%) | 0.296‡ |
| **Baseline cognitive function** | | | |
| Global Deterioration scale | 2.42 (0.50) | 2.54 (0.51) | 0.397* |
| K-MMSE | 21.71(1.33) | 21.92(1.21) | 0.574* |
| **baseline study variable** | | | |
| MoCA-K | 20 (17–21) | 19 (17–21) | 0.574# |
| GDS-KR | 8 (6–13) | 11 (9–17) | 0.580# |
| ADAS-K-cog-3 | 10 (7–11) | 10 (7–11) | 0.512# |
| K-ADL | 7 (7–8) | 7 (7–8) | 0.703# |
| K-IADL | 10 (10–12) | 11 (10–12) | 0.504# |
| EQ-5D-5L | 0.8 (0.8–0.8) | 0.8 (0.7–0.8) | 0.650# |

Values are expressed as means (standard deviation), medians (Q1– Q3) or counts (%)

*$p$ –value for the between-group comparison using an independent t-test.

‡$p$ –value for the between-group comparison using $x^2$-test

#$p$ –value for the between-group comparison using a Mann–Whitney U test

ADAS-K-cog 3, Alzheimer's Disease Assessment Scale-cognitive subscale-3; CWBSD, Cheonwangbosim-dan; EQ-5D-5L, European Quality of Life Five Dimension Five Level Scale; GDS-KR, Korean version of the Geriatric Depression Scale; K-ADL, Korean Activities of Daily Living; K-IADL, Korean Instrumental Activities of Daily Living; K-MMSE, Korean version of the the Mini-Mental State Examination; MoCA-K, Korean version of Montreal Cognitive Assessment scale.

**Table 2. Changes in MoCA-K scores at weeks 0, 12, and 24.**

| Variables | Group(n) | Week 0 | Week 12 | Week 24 | Difference (W 12–W 0) | *p*-value* | Difference (W 24–W 0) | *p*-value# effect size |
|---|---|---|---|---|---|---|---|---|
| MoCA-K | CWBSD (n = 24) | 19.21 (2.36) 20 (17–21) | 21.17 (3.87) 22 (18–24) | 22.58 (3.44) 23 (19–25) | 1.96 (2.66) | 0.597* | 3.38 (2.58) | 0.491# 0.234 |
| | Control (n = 24) | 18.83 (2.44) 19 (17–21) | 21.25 (3.25) 20 (19–24) | 22.88 (3.87) 23 (20–26) | 2.42 (3.27) | | 4.04 (3.03) | |

Values are expressed as mean (standard deviation) or medians (Q1- Q3)

*$p$ –value for the between-group comparison using the independent t-test.

#$p$ –value for the between-group comparison using a Mann–Whitney U test

MoCA-K, Korean version of Montreal Cognitive Assessment scale; CWBSD, Cheonwangbosim-dan.

with MCI. Our clinical trial revealed several findings. First, the CWBSD group did not exhibit a significant improvement in the total MoCA-K score compared to the control group. Second, there was no significant difference between the two groups in terms of GDS-KR, K-ADL, K-IADL, EQ-5D-5L, and ADAS-K-cog-3 scores. Third, CWBSD did not induce AEs/SAEs or significant negative changes in blood chemistry parameters, blood pressure, or pulse rate after treatment.

**Table 3. Changes in other secondary efficacy outcomes at weeks 0, 12, and 24.**

| Variables | Group(n) | Week 0 | Week 12 | Week 24 | Difference (W12–W0) | *p*-value | Difference (W24–W0) | *p*-value |
|---|---|---|---|---|---|---|---|---|
| GDS-KR | CWBSD (n=24) | 8 (6–13) | 7 (5–13) | 6 (6–11) | −0.42 (3.73) | 0.351# | −0.67 (3.24) | 0.645# |
| | Control (n=24) | 11 (9–17) | 7 (7–8) | 5 (3–11) | −1.00 (4.91) | | −1.75 (4.20) | |
| ADAS-K-cog-3 | CWBSD (n=24) | 10 (7–11) | 8 (6–10) | 7 (6–10) | −1.04 (3.06) | 0.700# | −2.21 (2.99) | 0.792# |
| | Control (n=24) | 10 (7–11) | 8 (6–10) | 7 (6–10) | −1.42 (2.78) | | −2.38 (3.12) | |
| K-ADL | CWBSD (n=24) | 7 (7–8) | 7 (7–7) | 7 (7–7) | −0.21 (0.59) | 0.907# | −0.08 (0.50) | 1.000# |
| | Control (n=24) | 7 (7–8) | 7 (7–7) | 7 (7–7) | −0.17 (0.38) | | −0.08 (0.50) | |
| K-IADL | CWBSD (n=24) | 10 (10–12) | 10 (10–11) | 10 (10–11) | −0.92 (1.61) | 0.555# | −0.88 (1.65) | 0.555# |
| | Control (n=24) | 11 (10–12) | 10 (10–11) | 10 (10–11) | −0.54 (1.22) | | −0.67 (1.31) | |
| EQ-5D-5L | CWBSD (n=24) | 0.8 (0.8–0.8) | 0.8 (0.8–0.8) | 0.8 (0.8–0.9) | 0.02 (0.05) | 0.531# | 0.02 (0.11) | 0.138# |
| | Control (n=24) | 0.8 (0.7–0.8) | 0.8 (0.8–0.8) | 0.8 (0.8–0.9) | 0.02 (0.07) | | 0.04 (0.05) | |

Values are expressed as medians (Q1–Q3)

#*p*–value for the between-group comparison using a Mann–Whitney *U* test.

ADAS-K-cog 3, Alzheimer's Disease Assessment Scale-cognitive subscale-3; CWBSD, Cheonwangbosim-dan; EQ-5D-5L, European Quality of Life Five Dimension Five Level scale; GDS-KR, Korean version of Geriatric Depression Scale; K-ADL, Korean Activities of Daily Living; K-IADL, Korean Instrumental Activities of Daily Living; K-MMSE, Korean version of Mini-Mental State Examination.

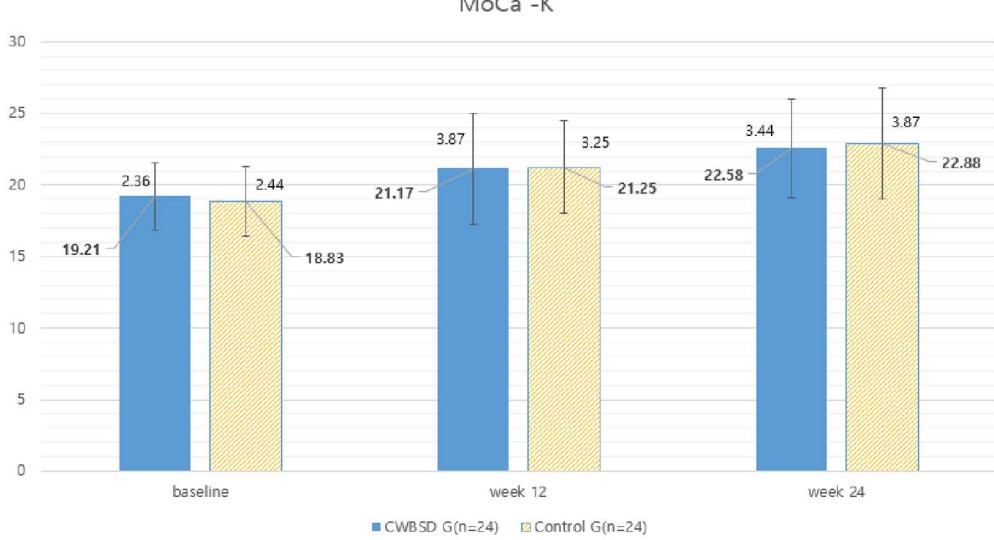

* indicates a significant difference between the CWBSD and control groups

**Fig 2. Changes in MoCA-K scores at weeks 0, 12, and 24.**

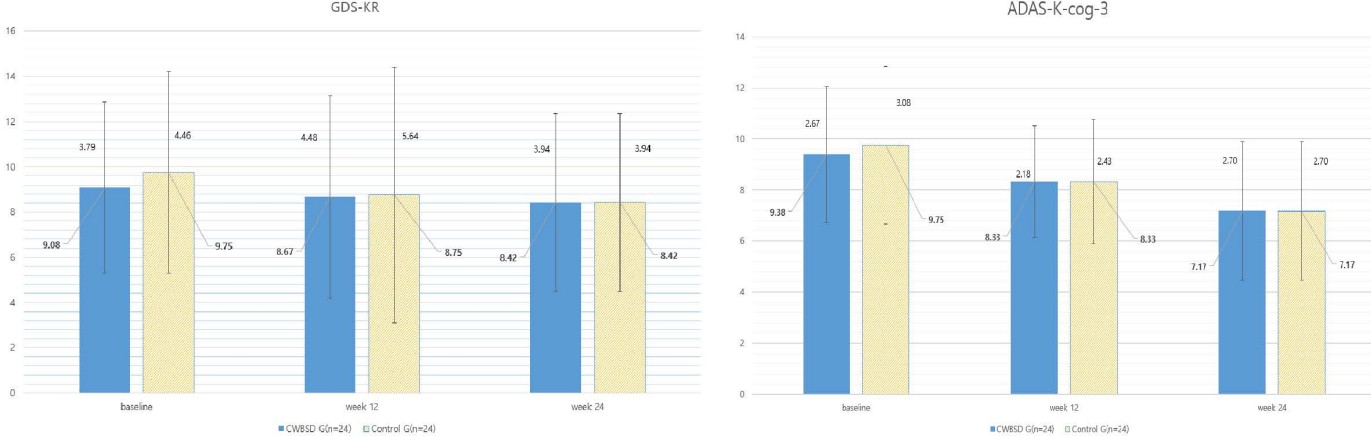

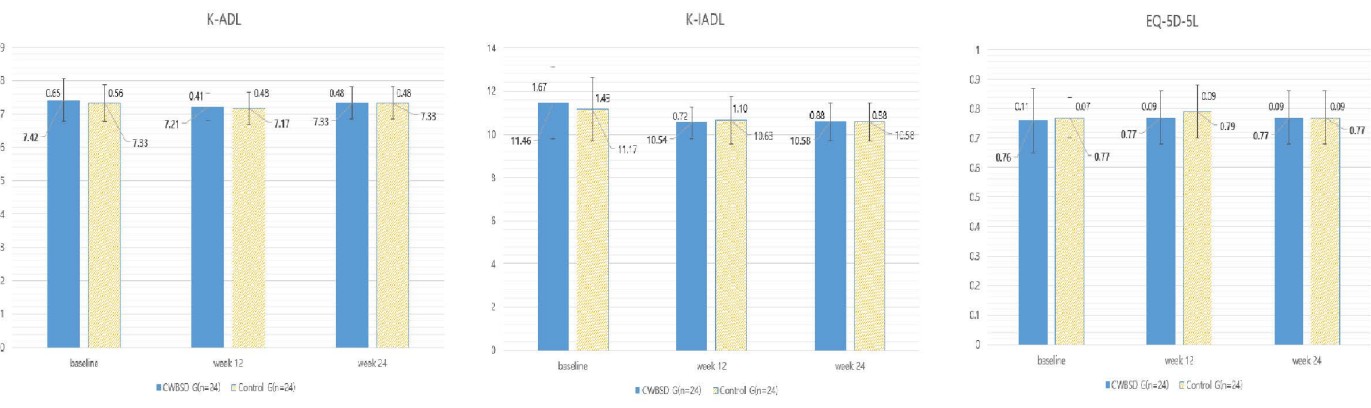

**Fig 3. Changes in secondary efficacy outcomes at weeks 0, 12, and 24.**

**Table 4. Incidence of AEs/SAE.**

| Variables | CWBSD (n = 24) | Control (n = 24) | *p*-value |
|---|---|---|---|
| Adverse events | Knee pain | Finger pain | – |
| Serious Adverse events | 1 (4.16%)<br>0 | 1 (4.16%)<br>0 | – |

Values are expressed as counts (%)

CWBSD, Cheonwangbosim-dan

**Table 5. Medication compliance at week 24.**

| Variables | CWBSD (n = 24) | Control (n = 24) | *p*-value |
|---|---|---|---|
| Medication compliance | 96.92 (93.55–99.84) | 96.92 (95.12–99.41) | 0.094# |

Values are expressed as medians (Q1– Q3)

#*p* –value for the between-group comparison using a Mann–Whitney U test.

CWBSD, Cheonwangbosim-dan

There are several possible explanations for this finding. First, the education on self-management and exercise provided to both groups may have influenced the results. Lifestyle modifications, including diet and physical activity, play a crucial role in managing MCI [40], and regular exercise is recommended in clinical practice guidelines to improve cognitive function in individuals with MCI [11]. However, because adherence to these lifestyle interventions was not actively monitored, it was not feasible to include them as covariates in our analysis. Future studies should systematically track diet and exercise adherence to assess their independent effects on cognitive outcomes. Second, the clinical decision-making process of traditional Chinese Medicine, including KM, may have affected the results. In KM, syndrome differentiation, also referred to as pattern identification, is the key decision-making process in clinical practice [41]. KM Clinicians diagnose patients' patterns after the professional syndrome differentiation process rather than at the disease stage and prescribe personalized herbal medicine to each patient based on KM syndrome differentiation. Previous studies have commonly reported patterns of yin deficiency, Qi deficiency, fire heat, blood stasis, and phlegm dampness among patients with cognitive impairments [42,43]. In a previous prospective observational study, approximately 80% of individuals with MCI were prescribed herbal formulae for yin (kidney) deficiency or Qi deficiency [44]. Yin (kidney) deficiency and Qi deficiency are the most common KM pattern types in patients with cognitive disorders. Thus, we adopted the CWBSD, which nourishes blood and enriches yin, as the intervention in this study. However, in older adults with MCI and phlegm dampness, blood stasis, and fire-heat KM pattern types, CWBSD may not affect cognitive function.

Our study had some limitations. First, the sample size was relatively small. Therefore, the sample size may not have been sufficient to investigate the efficacy of CWBSD in treating MCI. Therefore, the small sample size might have led to bias in the study results. Second, we did not consider syndrome differentiation based on the inclusion criteria. In a previous RCT that reported the efficacy of Qinggongshoutao for MCI, the inclusion criteria included a deficiency in kidney essence and adequate syndrome differentiation of Qinggongshoutao [14]. The inclusion criteria did not include adequate syndrome differentiation for CWBSD, which may have affected the results. Third, we investigated the short-term effects of CWBSD on MCI over a 24-week period. In previous studies, herbal medicines that exhibited significant effects on MCI in RCTs were administered for extended periods (e.g., 52 weeks [14], 2 years [15], and 24 months [45]). However, due to the short study period and limited research funding, we only investigated the short-term effects of CWBSD on MCI over a 24-week period in this study. Further studies are needed to examine the safety and long-term effects of CWBSD on MCI after 24 weeks.

## Conclusions

Our results suggest that a 24-week course of CWBSD treatment was well tolerated, with no observed drug-related adverse events or significant drug-induced alterations in blood pressure, pulse rate, or blood chemistry parameters. However, CWBSD did not demonstrate a significant effect on improving cognitive function, quality of life, daily activities, or depression in individuals with MCI. Further studies with longer treatment durations and larger sample sizes are needed to fully assess the long-term safety and efficacy of CWBSD in MCI management.

## Supporting information

**S1 File. Data.**
(XLSX)

**S2 File. CONSORT checklist.**
(DOCX)

**S3 File. Study protocol.**
(PDF)

**S4 File. SPIRIT statement.**
(DOCX)

**S5 File. Components appearance, and packaging of CWBSD.**
(DOCX)

**S6 File. The comparison of blood chemistry test parameters.**
(DOCX)

## Acknowledgments

The authors express their sincere thanks to their colleagues and the staff at Dongshin University Gwangju Korean Medicine Hospital for their support.

## Author contributions

**Conceptualization:** Jae-Hong Kim.

**Data curation:** Jae-Hong Kim, Jeong-Soon Lee.

**Formal analysis:** Jeong-Soon Lee.

**Funding acquisition:** Jae-Hong Kim.

**Investigation:** Jae-Hong Kim, Gwang-Cheon Park.

**Methodology:** Ae-Ran Kim, Bok-Nam Seo.

**Project administration:** Jae-Hong Kim.

**Resources:** Ae-Ran Kim, Bok-Nam Seo, Gwang-Cheon Park.

**Software:** Gwang-Cheon Park.

**Supervision:** Jae-Hong Kim.

**Validation:** Jae-Hong Kim.

**Visualization:** Ae-Ran Kim, Bok-Nam Seo, Gwang-Cheon Park.

**Writing – original draft:** Jae-Hong Kim.

**Writing – review & editing:** Jae-Hong Kim, Ae-Ran Kim, Bok-Nam Seo, Gwang-Cheon Park, Jeong-Soon Lee.

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
