## [Editor Report · Decision Letter 0]

PONE-D-23-33277The efficacy and safety of Cheonwangbosim-dan (Tian Wang Bu Xin Dan) for treatment of mild cognitive impairment: a randomized placebo-controlled trialPLOS ONE

Dear Dr. Kim,

Thank you for submitting your manuscript to PLOS ONE. After careful consideration, we feel that it has merit but does not fully meet PLOS ONE’s publication criteria as it currently stands. Therefore, we invite you to submit a revised version of the manuscript that addresses the points raised during the review process.

This is a good study. There is novelty in this topic. 

Please give some details and improvement in this part:

- Table 1 and table 2, I do not think is necessary. The authors can give it as supplements.

- Please consider any co-morbidities and co-medication in baseline characteristic. 

- Please follow the CONSORT guideline. 

- Please consider any confounders in co-medication. 

- The graph should be used to describe the improvement. 

- The discussion should be begin with the main finding of your study. 

- The conclusion should be concise and clear.

We look forward to receiving your revised manuscript.

Kind regards,

Rizaldy Taslim Pinzon

Academic Editor

PLOS ONE

Journal Requirements:

3. We are unable to open your Supporting Information file "plos one-S2 file-CONSORT 2010 Checklist.doc". Please kindly revise as necessary and re-upload.

Additional Editor Comments:

This is a good study.

There is novelty in this topic.

Please give some details and improvement in this part:

- Table 1 and table 2, I do not think is necessary. The authors can give it as supplements.

- Please consider any co-morbidities and co-medication in baseline characteristic.

- Please follow the CONSORT guideline.

- Please consider any confounders in co-medication.

- The graph should be used to describe the improvement.

- The discussion should be begin with the main finding of your study.

- The conclusion should be concise and clear.

---

## [Author Response · Author response to Decision Letter 1]

29 Nov 2023

Dear Editor,

We are immensely grateful for the constructive comments and the opportunity to revise our manuscript. Point-by-point responses to the journal requirements and the reviewers’ comments are provided below. The corresponding changes have been highlighted in the revised manuscript with tracked changes.

Additional Editor Comments:

This is a good study.

There is novelty in this topic.

Please give some details and improvement in this part:

- Table 1 and table 2, I do not think is necessary. The authors can give it as supplements.

Response. We sincerely appreciate your encouraging and uplifting comments. As per your invaluable suggestion, we have provided tables 1 and 2 as S4 and S5 files, respectively.

- Please consider any co-morbidities and co-medication in baseline characteristic.

Response. We appreciate your invaluable suggestion. In a prior study [1], modifiable risk factors for MCI were identified, including smoking, hypertension, diabetes, hyperlipidemia, heart disease, and cerebrovascular diseases. Consequently, we focused on examining the homogeneity of baseline characteristics, encompassing these comorbidities as well as concomitant medication usage. Furthermore, table 1 has been updated to incorporate these factors, namely smoking, hypertension, diabetes, hyperlipidemia, heart disease, and cerebrovascular disease, as dependent variables.

- Please follow the CONSORT guideline.

Response. We have attached the CONSORT 2010 checklist as the S2 file in the supporting information.

- Please consider any confounders in co-medication.

Response. Thank you for bringing this up. We have performed a homogeneity test to compare the use of medication for hypertension, diabetes, hyperlipidemia, heart disease, and cerebrovascular disease between the two groups, as detailed in table 1. The results indicated no significant differences in medication usage for these conditions at baseline between the two groups. Consequently, we determined that it was unnecessary to perform an ANCOVA with co-medication as a covariate.

- The graph should be used to describe the improvement.

Response. In accordance with your recommendation, we have presented the changes in the MoCA-K scores in figure 2. Additionally, Figures 3 and 4 present the results of the sub-analysis stratified by age. We have also illustrated the variations in secondary efficacy outcomes in figure 5, with the age-stratified sub-analysis results further detailed in figures 6, 7.

- The discussion should be begin with the main finding of your study.

Response. As per your invaluable recommendation, we have revised the discussion section to begin with the main findings of our study as follows: “ Discussion

The main findings of our study reveal that CBD demonstrated an accepable safety profile and exhibited a significant effect in short-term memory functions. However, it failed to show significant improvements in other cognitive domains, quality of life, daily activities, or depression among patients with MCI. (Page 18, lines 348–351)”

- The conclusion should be concise and clear.

Response. We appreciate your insightful comment. In accordance with your suggestion, we have revised the conclusion section to read: “Our findings indicate that a 24-week treatment regimen with CBD was safe and significantly enahnced short-term memory in patients with MCI. Further rigorously designed clinical studies are pertinent to validate the efficacy of CBD in the managemnt of MCI.” (Page 20, line 408-410)

Reference

1. Jia L, Du Y, Chu L, Zhang Z, Li F, Lyu D, et al. Prevalence, risk factors, and management of dementia and mild cognitive impairment in adults aged 60 years or older in China: A cross-sectional study. Lancet Public Health. 2020;5: e661-e671. doi: 10.1016/S2468-2667(20)30185-7.

---

## [Decision Letter · Decision Letter 1]

PONE-D-23-33277R1The efficacy and safety of Cheonwangbosim-dan (Tian Wang Bu Xin Dan) for treatment of mild cognitive impairment: a randomized placebo-controlled trialPLOS ONE

Dear Dr. Kim, 

Thank you for submitting your manuscript to PLOS ONE. After careful consideration, we feel that it has merit but does not fully meet PLOS ONE’s publication criteria as it currently stands. Therefore, we invite you to submit a revised version of the manuscript that addresses the points raised during the review process.

We look forward to receiving your revised manuscript.

Kind regards,

Rizaldy Taslim Pinzon

Academic Editor

PLOS ONE

**Additional Editor Comments:**

Thank you authors for the prompt reply.

Please follow the CONSORT guideline.

I suggest some points of improvements based on the comments from the reviewers.

You can find detail below.

Reviewers' comments:

Reviewer's Responses to Questions

**Comments to the Author**

1. If the authors have adequately addressed your comments raised in a previous round of review and you feel that this manuscript is now acceptable for publication, you may indicate that here to bypass the “Comments to the Author” section, enter your conflict of interest statement in the “Confidential to Editor” section, and submit your "Accept" recommendation.

Reviewer #1: (No Response)

Reviewer #2: (No Response)

2. Is the manuscript technically sound, and do the data support the conclusions?

Reviewer #1: Partly

Reviewer #2: No

3. Has the statistical analysis been performed appropriately and rigorously? 

Reviewer #1: No

Reviewer #2: No

4. Have the authors made all data underlying the findings in their manuscript fully available?

Reviewer #1: Yes

Reviewer #2: No

5. Is the manuscript presented in an intelligible fashion and written in standard English?

Reviewer #1: Yes

Reviewer #2: No

6. Review Comments to the Author

Reviewer #1: The manuscript can be further improved based on the comments below.

Line 152, who are the outcome assessors is to be stated.

Line 156-158, the sentence to be placed in Line 151.

Line 205, whether the effect size 1.252, was derived from the intervention group alone or after comparison with the control group is to be clearly stated.

Line 212, the justification to use LOCF is to be mentioned e.g. MCAR etc.

Line 223, one or two-tailed test for Fisher’s exact test is to be stated.

Line 234 & Line 236, PPS or PP?

For inventories such as GDS, EQ-5D-5L, whether Korean version was used is to be clearly stated. Information on the validity/reliability of the instruments used in the study should be briefly described.

Line 259, the point of analysis e.g. week 12 and week 24 is to be stated. The table to be cited.

Table 1, the column p or X^2(p) is confusing. p is to be denoted in the table footnote. Suggest to use just p-value. Some of the p-value symbols are incorrectly labeled. Please re-check the statistical analysis. Denote which one is for the chi-square test and which one is for Fisher’s exact test. The decimal point for the percentage figure is to be consistent. If Fisher’s exact test was not utilized (based on Line 248), the statistical test is to be omitted from mention in the statistical analysis section in the methodology.

For Table 1, having said that, based on CONSORT statement all statistical tests for baseline comparison are to be avoided. The choice to adjust the required variables in the analysis from the baseline characteristics could be based on either clinical relevance, known confounders, visual inspection, sensitivity analyses etc.

Table 2, effect size indices could be presented. The table requires cosmetic changes e.g. alignment of the figures. This applies to all other tables.

Line 281, the data in Table 2 are median and IQR. As such, the word mean (sd) in Line 281 is to be removed.

ANCOVA is a parametric test with assumptions of normality. Are those skewed data transformed before analysis? If not transformed, how skewed are the data? More information is to be provided in the methodology section.

Line 288, if nonparametric tests are used, median and IQR are to be displayed. Likewise, with Table 4 Line 295, Table 5 Line 308, Table 6 Line 318, and Table 7 Line 328.

Figure 1, revision required for the analysis section. Excluded and per protocol analysis number is to be separated.

Per protocol analysis is to be highlighted or denoted where necessary in the table(s) and/or figure(s).

For Figure 2-7, n is to be stated for the CBD group and control group.

The statement ‘*indicates a significant difference between the CBD and control groups’ for Figure 2-7 is to be denoted in the figures.

Reference 35, the DOI link inaccessible https://doi.org/10.3736/jcim20120407

Please ensure all links provided in the reference list are accessible. The spacing for some references is inconsistent.

Reviewer #2: Thank you for the opportunity to review this paper. Whilst this is a well-designed and executed study, the analyses do not appear to be valid. There is no statistically significant group x time interaction (or equivalent) on the primary outcome – the global cognitive screener, MoCA-K. This is OK and should be reported as a null result. However, MoCA-K is divided into sub-tests and these are all analysed and reported without controlling for type 1 error. The relatively small sample is then further stratified into age <70 and >70, which the study was not powered to do and there is no strong argument for the protocol deviation. I have left specific comments section-by-section below.

Introduction:

MCI is repeatedly discussed throughout in the context of AD. However, around half of people with MCI do not progress to dementia. Further, only around 45% of people with MCI who progress to dementia develop the AD sub-type. Unless MCI is specifically being characterised as prodromal AD (either with clinical or pathological phenotypes), then this direct assertion to AD should be modified. Instead, MCI should be discussed as a dementia prodrome or dementia risk factor.

I wonder about the acronym of CBD for the trial IMP. CBD is a commonly used abbreviation for cannabidiol and this may confuse potential readers.

Methods:

Why did the intervention also include education ‘on exercise and self-management’? Would this not potentially confound the intervention? What did the education component look like and how was it delivered?

Regarding the eligibility criteria. There is no set of diagnostic criteria for MCI in the DSM-V. Instead, it refers to major and mild neurocognitive disorder. Please describe how this was operationalised for the current study.

What is the MCI range for the MoCA-K? The English version uses scores 18 – 25 for MCI, so would 0–22 on the MoCA-K also include dementia?

Regarding dropout criteria – please provide definition of SAE used or refer to the safety reporting guidelines so that the standardised criteria can be viewed.

What exactly is contained within the CBD active? Is this a herbal medicine formulation? If so, which herbs? Are they standardised extracts? More information is needed here.

What time of day did the once-daily administration of the IMP occur?

For the sample size calculation, what unit of measurement is the expected effect size (1.252) reported in? Cohen’s d?

Which outcome measure was used to determine the sample size calculation?

It’s not clear in the definition of primary and secondary outcome measures that sub-scales from within the MoCA-K will be reported. Was this part of the initial analysis plan, or was it a modification of the protocol? Is it valid to analyse sub-scales from the MoCA-K? The English version is typically reported as the global score (/30) and sub-scales are not analysed due to lack of validity. Re-analysing sub-scales from the MoCA also increases the probability of type 1 error – was testing for multiple comparisons controlled for?

The sub-group analysis by age (70) is also likely to be underpowered to detect anything given that this was not accounted for the original sample size calculation. It looks as though each variable has been ‘sliced-up’ multiple times, and now the probability for type 1 error is very high.

Results:

In text, dropouts are attributed to ‘asked to leave the trial’, but in the CONSORT flow diagram (Figure 1), participants withdrew their consent. Please provide more detail here on the reason for consent withdrawal and/or why participants were asked to leave the trial.

The figures could be improved by placing together in a single image with multiple panels. There should also be standard error/standard deviation bars.

Were variables checked for normality? Did any log-normalisation occur?

It’s not clear which timepoints are being compared from the within group comparisons (baseline vs. midpoint, midpoint vs. endpoint, baseline vs. endpoint). I’m also not sure why within-group comparisons are conducted and reported as this was not the primary objective to the trial.

Compliance data (returned % of unused medication) is not reported. Full details on safety AEs/SAEs should be tabulated.

Other:

“elderly” is an ageist label. Please see one of the many international guidelines on age-positive language and utilise this throughout (e.g., older people, older adults etc.).

The manuscript would benefit from high-level editing and polishing by a native English speaker.

Tables are difficult to read as the text is split across multiple lines for the same variable (e.g., see ‘Total’ row in Table 2).

7. PLOS authors have the option to publish the peer review history of their article (what does this mean? ). If published, this will include your full peer review and any attached files.

**Do you want your identity to be public for this peer review?** For information about this choice, including consent withdrawal, please see our Privacy Policy .

Reviewer #1: No

Reviewer #2: **Yes: ** Genevieve Z. Steiner-Lim

---

## [Author Response · Author response to Decision Letter 2]

16 May 2024

We are very grateful for the constructive comments and the opportunity to revise our manuscript.

Point-by-point responses to the reviewers’ comments are provided below. The corresponding changes have been highlighted in the revised manuscript with track changes.

Question: Reviewer #1: The manuscript can be further improved based on the comments below.

Line 152, who are the outcome assessors is to be stated.

Response: Thank you for your comments. Outcome assessors are those who evaluate the efficacy outcome, including MoCA-K, ADAS-K-cog 3, GDS-KR, EQ-5D-5L, K-ADL, and K-IADL, as well as safety outcomes. They were received training for the assessment tools employed in our study. We have provided a more detailed description of the outcome assessors as follows: “Throughout the trial, all participants, outcome assessors who were received training for the assessment tools employed in our study and evaluated the efficacy and safety outcomes, and the investigators remained blinded to the group assignments.” (Page 7, lines 156–158 )

Question: Line 156-158, the sentence to be placed in Line 151.

Response: Thank you for your valuable comments. Following your suggestion, we have revised the randomization and blinding subsection as follows: “The randomization numbers were sealed in opaque envelopes and stored in a double-locked cabinet. An independent investigator, who was not involved in the trial, was responsible for generating the allocation sequence, whereas a clinical research coordinator managed the participant enrollment and group assignments.

Throughout the trial, all participants, outcome assessors who were received training for the assessment tools employed in our study and evaluated the efficacy and safety outcomes, and the investigators remained blinded to the group assignments.” (Page 7, line 152-158)

Question: Line 205, whether the effect size 1.252, was derived from the intervention group alone or after comparison with the control group is to be clearly stated.

Response: Thank you for your valuable comments. The effect size was calculated based on the comparison between the intervention group and the control group. As there were no existing data demonstrating the efficacy of CWBSD using MoCA-K for sample size calculations, we opted for a pilot study design. The sample size was determined based on a previous randomized controlled trial (RCT) investigating the effects of Chinese herbal medicine using MMSE and ADAS-cog in patients with MCI. The effect size (1.252) was obtained from the differences in pre- and post-treatment scores of ADAS-cog between the Chinese herbal medicine and control group in the referenced RCT.

Since access to the aforementioned article was not available, we have revised the sample size estimation subsection as follows: “We could not find any existing data showing the efficacy of CWBSD using MoCA-K, which is required for sample-size calculations. Therefore, we adopted a pilot-scale study design. The appropriate sample size for the pilot study was >12 [37]. As our study was a pilot trial, formal sample size estimation was not conducted. Instead, considering the sample size of a previous pilot study that explored the efficacy and safety of Chinese herbal medicine for MCI [16], 48 participants were recruited and equally distributed, with 24 participants assigned to each group. The sample size may be insufficient to determine the effects of CWBSD on MCI. The findings of our study provide preliminary data on the effects of CWBSD on MCI.” (Page 10, line 222–230)

Question: Line 212, the justification to use LOCF is to be mentioned e.g. MCAR etc.

Response: Thank you for your valuable comments. Last observation carried forward is a common statistical approach to analysis of longitudinal repeated measures data where some follow-up observations may be missing. Following your suggestion, we have revised the statistical analyses subsection as follows: “The efficacy evaluation was performed using a full analysis set. Because the missing variable was only affected by random factors (missing completely at random), missing values were imputed using the last observation carried forward technique. (Page 11, line 233–236)

Question: Line 223, one or two-tailed test for Fisher’s exact test is to be stated.

Response: Thank you for your comments. Since we did not employ Fisher’s exact test in our statistical analyses, it was omitted from mention in the statistical analysis section in the methodology.

Question: Line 234 & Line 236, PPS or PP?

Response: We sincerely regret any confusion. The mention of the per protocol set in our manuscript was an error on our part. In the original protocol, the analysis of treatment efficacy was intended to be conducted on the full analysis set with a supplementary per protocol set. However, we have since revised our statistical analysis methods and obtained IRB approval to exclude the supplementary per protocol set. Consequently, we have removed all content related to the per protocol set from the manuscript.

Question: For inventories such as GDS, EQ-5D-5L, whether Korean version was used is to be clearly stated. Information on the validity/reliability of the instruments used in the study should be briefly described.

Response: Thank you for your valuable comments. The GDS and EQ-5D-5L utilized in our study are the Korean versions of these questionnaires. We have provided information on the validity and reliability of the instruments used in the study in the efficacy outcome subsection as follows: “The Montreal Cognitive Assessment Scale is a commonly used instrument for screening MCI in older individuals, measuring cognitive functions across various domains. It is a 30-point test evaluating the cognitive domains of executive function, concentration, attention, working memory, short-term memory recall, visuospatial abilities, orientation to place and time, and language [28]. The MoCA-K is a brief, suitable, and reliable tool for screening MCI patients. The optimal cut-off score of the MoCA-K (22/23) is three points lower than the original MoCA, which may be due to low education years and cultural differences [29].

The Geriatric Depression Scale is a widely used self-report screening instrument to assess depressive symptoms in older adults. It was specifically designed to differentiate between the symptoms of depression and dementia in this demographic. The GDS-KR is a valid and reliable questionnaire for screening major and minor depressive disorders in older people [30].

The Korean version of the Alzheimer’s Disease Assessment Scale is a reliable and valid tool for diagnosing AD and evaluating its severity [31]. The ADAS-K-cog 3, comprising orientation, word recall, and word recognition tasks from the ADAS-K-cog 11, is particularly effective for MCI as it does not exhibit ceiling effects. This tool focuses exclusively on memory assessment [32].

The EQ-5D-5L is a widely used patient-reported health utility instrument for evaluating the quality of life. The Korean version of the EQ-5D-5L is a reliable and valid tool for measuring health-related quality of life in the general population of South Korea [33]. The quality weight for the EQ-5D-5L was based on previous studies [34].

K-IADL and K-ADL are valid and reliable instruments for assessing basic and complex daily activities in older people [35,36]. (Page 9–10, line 193–214)

Question: Line 259, the point of analysis e.g. week 12 and week 24 is to be stated. The table to be cited.

Response: Thank you for your comments. Following your suggestion, we have revised the efficacy evaluation subsection as follows: “We compared the degrees of change in the MoCA-K, GDS-KR, K-ADL, K-IADL, ADAS-K-cog-3, and EQ-5D-5L scores between the two groups (Week 12 vs. Week 0 and Week 24 vs. Week 0) using the independent t-test or Mann–Whitney U test. (Tables 2 and 3)” (Page 13, line 280–282)

Question: Table 1, the column p or X^2(p) is confusing. p is to be denoted in the table footnote. Suggest to use just p-value. Some of the p-value symbols are incorrectly labeled. Please re-check the statistical analysis. Denote which one is for the chi-square test and which one is for Fisher’s exact test. The decimal point for the percentage figure is to be consistent. If Fisher’s exact test was not utilized (based on Line 248), the statistical test is to be omitted from mention in the statistical analysis section in the methodology.

Response: We sincerely regret for causing confusion. Following your suggestion, we have revised Table 1. In Table 1, we have used only p-values, corrected the symbols for p-values, and consistently revised the decimal point for percentage figures. Since we did not utilize Fisher’s exact test in our statistical analyses, it was omitted from mention in the statistical analysis section in the methodology.

Question: For Table 1, having said that, based on CONSORT statement all statistical tests for baseline comparison are to be avoided. The choice to adjust the required variables in the analysis from the baseline characteristics could be based on either clinical relevance, known confounders, visual inspection, sensitivity analyses etc.

Response: Thank you for your valuable comments. In a prior study [1], the prevalence of MCI was associated with age, sex, years of education, comorbidities such as hypertension, hyperlipidemia, and diabetes, and other factors. In accordance with your suggestion, in Table 1, we compared the demographic characteristics, including age, gender, years of education, smoking status, and comorbidities (specifically, hypertension, diabetes mellitus, dyslipidemia, and heart disease), as well as baseline cognitive function (measured by global deterioration and K-MMSE), and baseline study variables (measured by MoCA-K, GDS-KR, ADAS-K-cog3, K-ADL, K-IADL, and EQ-5D-5L).

Question: Table 2, effect size indices could be presented. The table requires cosmetic changes e.g. alignment of the figures. This applies to all other tables.

Response: Thank you for your valuable comments. Following your suggestions, we have included the effect size indices in Table 2 and revised all tables in our manuscript to enhance the visibility of the content.

Question: Line 281, the data in Table 2 are median and IQR. As such, the word mean (sd) in Line 281 is to be removed.

Response: Thank you for your comments. Following the suggestion of reviewer 2 and after consulting with a statistician not involved in this study, as well as reaching a consensus among the researchers, we have decided to remove the contents related to the analyses on the MoCA-K subscale and sub-analyses by age. Consequently, we have removed Tables 3, 4, 6, 7, Figures 3, 4, 6, 7, and the contents of the MoCA-K subscale from Table 2.

We conducted a Shapiro-Wilk test to assess the normality of data distribution. While the difference (W12-W0) demonstrated a normal distribution, the difference (W24–W0) did not. Therefore, we have presented the data as mean (standard deviation) and median (Q1–Q3) and revised it as follows: “Values are expressed as mean (standard deviation) or medians (Q1–Q3)” (Page 13, line 289)

Question: ANCOVA is a parametric test with assumptions of normality. Are those skewed data transformed before analysis? If not transformed, how skewed are the data? More information is to be provided in the methodology section.

Response: We apologize for the confusion. It is our mistake. We conducted a Shapiro-Wilk test to assess the normality of data distribution. Except for the difference (Week 12 vs. Week 0) in MoCA-K, all efficacy outcomes did not exhibit a normal distribution. Consequently, we have removed the contents related to ANCOVA from the manuscript.

Question: Line 288, if nonparametric tests are used, median and IQR are to be displayed. Likewise, with Table 4 Line 295, Table 5 Line 308, Table 6 Line 318, and Table 7 Line 328.

Response: Thank you for your valuable comments. Following the suggestion of reviewer 2 and after consulting with a statistician not involved in this study, as well as reaching a consensus among the researchers, we have removed the contents related to the analyses on the MoCA-K subscale and sub-analyses by age. This includes removing Table 3, 4, 6, 7, Figures 3, 4, 6, 7, and the contents of the MoCA-K subscale from Table 2.

We conducted a Shapiro-Wilk test to assess the normality of data distribution. We have revised the footnotes of Tables 2 and 3 as follows: Table 2 “Values are expressed as mean (standard deviation) or medians (Q1-Q3)

* p –value for the between-group comparison using independent t-test.

#p –value for the between-group comparison using a Mann–Whitney U test” (Page 14, line 289-291)

Table 3 “Values are expressed as medians (Q1, Q3)

#p –value for the between-group comparison using a Mann–Whitney U test” (Page 15, line 300–301)

Question: Figure 1, revision required for the analysis section. Excluded and per protocol analysis number is to be separated.

Response: We extend our apologies for any confusion that may have arisen. The mention of the per protocol set in Figure 1 was our mistake. In the original protocol, the analysis of treatment efficacy was intended to be conducted on the full analysis set with a supplementary per protocol set. We planned to compare the results of analyses between the full analysis set and the per protocol set to confirm whether there were statistically significant differences between the two groups. However, we have since revised our statistical analysis methods with IRB approval to exclude the supplementary per protocol set. Consequently, we have removed all content related to the per protocol set from Figure 1.

Question: Per protocol analysis is to be highlighted or denoted where necessary in the table(s) and/or figure(s).

Response: We extend our apologies for any confusion that may have arisen. The mention of the per protocol set in our manuscript was our mistake. In the original protocol, the analysis of treatment efficacy was planned to be conducted on the full analysis set with a supplementary per protocol set. The intention was to compare the results of analyses between the full analysis set and the per protocol set to confirm whether there were statistically significant differences between the two groups. However, we have since revised our statistical analysis methods with IRB approval to exclude the supplementary per protocol set. Consequently, we have removed all content related to the per protocol set from the manuscript.

Question: For Figure 2-7, n is to be stated for the CBD group and control group.

Response: Thank you for your comments. Following the suggestion of reviewer 2 and after consulting with a statistician not involved in this study, as well as reaching a consensus among the researchers, we have removed the contents related to the analyses on the MoCA-K subscale and sub-analyses by age. This includes removing Figures 3, 4, 6, 7, and the contents of the MoCA-K subscale from Figure 2. We have also described the sample size (n) for the CWBSD and control groups in Figures 2 and 3.

Question: The statement * indicates a significant difference between the CBD and control groups for Figure 2-7 is to be denoted in the figures.

Response: Thank you for your comments. Following the suggestion of reviewer 2 and after consulting with a statistician not involved in this study, as well as reaching a consensus among the researchers, we have removed the contents related to the analyses on the MoCA-K subscale and sub-analyses by age. This includes removing Figures 3, 4, 6, 7, and the contents of the MoCA-K subscale from Figure 2.

We have inserted the statement '*indicates a significant difference between CWBSD and control groups' in Figures 2 and 3.

Question: Reference 35, the DOI link inaccessible https://doi.org/10.3736/jcim20120407

Response: Thank you for your valuable comments. Since we do not have access to the aforementioned article, we have revised the sample size estimation subsection as follows: “We could not find any existing data showing the efficacy of CWBSD using MoCA-K, which is required for sample-size calculations. Therefore, we adopted a pilot-scale study design. The appropriate sample size for the pilot study was >12 [37

---

## [Decision Letter · Decision Letter 2]

PONE-D-23-33277R2The efficacy and safety of Cheonwangbosim-dan (Tian Wang Bu Xin Dan) for treatment of mild cognitive impairment : a randomized placebo-controlled pilot trialPLOS ONE

Dear Dr. Kim,

Thank you for submitting your manuscript to PLOS ONE. After careful consideration, we feel that it has merit but does not fully meet PLOS ONE’s publication criteria as it currently stands. Therefore, we invite you to submit a revised version of the manuscript that addresses the points raised during the review process.

We look forward to receiving your revised manuscript.

Kind regards,

Muhammad Salman Bashir, M.S.C

Academic Editor

PLOS ONE

Reviewers' comments:

Reviewer's Responses to Questions

**Comments to the Author**

1. If the authors have adequately addressed your comments raised in a previous round of review and you feel that this manuscript is now acceptable for publication, you may indicate that here to bypass the “Comments to the Author” section, enter your conflict of interest statement in the “Confidential to Editor” section, and submit your "Accept" recommendation.

Reviewer #1: All comments have been addressed

Reviewer #3: (No Response)

Reviewer #4: (No Response)

2. Is the manuscript technically sound, and do the data support the conclusions?

Reviewer #1: Partly

Reviewer #3: Partly

Reviewer #4: Partly

3. Has the statistical analysis been performed appropriately and rigorously? 

Reviewer #1: No

Reviewer #3: No

Reviewer #4: I Don't Know

4. Have the authors made all data underlying the findings in their manuscript fully available?

Reviewer #1: (No Response)

Reviewer #3: No

Reviewer #4: No

5. Is the manuscript presented in an intelligible fashion and written in standard English?

Reviewer #1: Yes

Reviewer #3: Yes

Reviewer #4: Yes

6. Review Comments to the Author

Reviewer #1: The authors have addressed the comments.

Minor comment

The symbol * and ǂ from the ‘p value’ in Table 3 are to be omitted.

Reviewer #3: In the present study,Kim et.al have investigated the safety and efficacy of

Cheonwangbosim-dan (CWBSD) for treating mild cognitive impairment (MCI). They observed that that CWBSD treatment was safe but did not significantly improve cognitive function in patients with MCI.

The study was conducted as a double-blind placebo-controlled design which is robust.

The study has several limitations. The conclusion is not clear.

Abstract-

The dose of CWBSD can be mentioned in the abstract.

Introduction:

• MCI relates to a state when the cognitive ability is not as good as it used to be. Although it refers to problems affecting memory, but could involve a change in problem solving, thinking, attention, concentration, language or visual ability. This needs to be clearly explained in the introduction.

• MCI is a mild state of cognitive impairment. It is not a disease. It may not be correct to refer to individuals with MCI as “patients”.

• What is CWBSD? It needs to be briefly explained in the introduction.

Methods

• How was 3g dose of CWBSD administered? Was it a capsule/tablet ? divided dose? This needs to be elaborated

• Physical exercise can have a profound impact on mental function. What kind of exercise was included in the study?

• It is also well known that food habits affect mental performance. The diet and lifestyle management need to be elaborated.

• In a 24-week study, how was compliance ensured? Were the participants contacted at regular intervals to remind them to take the supplements?

• How was safety ensured since the participants visited the hospital only after 12 weeks.

Results

• At what stage did the participants drop out of the trial. Only 38 participants completed the study. The numbers used for analysis is very small. How could the data for 48 participants be used for efficacy analysis?

• Safety data on biochemistry and hematology should be included at least as a supplementary table

• In 24 weeks only 2 participants reported an adverse event? Were all events captured?

Discussion

• There are several limitations in the study. Although age was significantly higher in CWBSD, all the baseline scores of outcome parameters were almost the same. So this could not be a reason for t CWBSD group not exhibiting a significant improvement.

• The exercise and diet could have had an effect in both groups. It would have made an impact to correlate the effect of these factors on cognitive improvement. May be use them as covariate for the analysis.

• A dose of 3g/day is high. Increasing the dose can have other deleterious effects

• The conclusion sentence in incomplete

Reviewer #4: Due to the small sample size of this study, there are limitations in evaluating efficacy. Therefore, the safety evaluation is of particular importance. In this trial, only two adverse events (AEs) were reported, which appears to be a lower frequency compared to other clinical trials. Could you explain why the number of AEs was so low? Specifically, please provide justification for how AEs were adequately captured in this trial. Additionally, please provide a statistical description of the elevated potassium concentrations in the CWBSD group mentioned in the text, including standard deviation and interquartile range.

7. PLOS authors have the option to publish the peer review history of their article (what does this mean? ). If published, this will include your full peer review and any attached files.

**Do you want your identity to be public for this peer review?** For information about this choice, including consent withdrawal, please see our Privacy Policy .

Reviewer #1: No

Reviewer #3: No

Reviewer #4: No

---

## [Author Response · Author response to Decision Letter 3]

14 Mar 2025

We are very grateful for the constructive comments and the opportunity to revise our manuscript.

Point-by-point responses to the reviewers’ comments are provided below. The corresponding changes have been highlighted in the revised manuscript with track changes.

Reviewer #1

Comment: The authors have addressed the comments.

Minor comment

The symbol * and ǂ from the ‘p value’ in Table 3 are to be omitted.

Response: Thank you for your valuable comments. After considering your suggestion, we have omitted the symbol * and ǂ from the ‘p value’ in Table 3.

Reviewer #3:

In the present study, Kim et.al have investigated the safety and efficacy of

Cheonwangbosim-dan (CWBSD) for treating mild cognitive impairment (MCI). They observed that that CWBSD treatment was safe but did not significantly improve cognitive function in patients with MCI. The study was conducted as a double-blind placebo-controlled design which is robust. The study has several limitations. The conclusion is not clear.

Comment: Abstract: The dose of CWBSD can be mentioned in the abstract.

Response: Thank you for your valuable comment. As per your suggestion, we have included the dose of CWBSD in the abstract to enhance clarity. The revised abstract now states: “The participants were administered either placebo or CWBSD once daily (at a dose of 3 g) for twenty-four weeks.” (Page 2, line 30–31)

Comment: Introduction:

MCI relates to a state when the cognitive ability is not as good as it used to be. Although it refers to problems affecting memory, but could involve a change in problem solving, thinking, attention, concentration, language or visual ability. This needs to be clearly explained in the introduction.

Response: Thank you for your valuable comments. As per your suggestion, we have revised the introduction to provide a clearer explanation of mild cognitive impairment (MCI). The revised section now states: “Mild cognitive impairment (MCI) is an intermediate stage between normal cognitive aging and dementia. It is characterized by a decline in one or more cognitive domains, including memory, problem-solving, attention, concentration, language, executive functions, or visuospatial abilities, while daily activities remain largely intact [1, 2].” (Page 3, line 50–53)

Comment: MCI is a mild state of cognitive impairment. It is not a disease. It may not be correct to refer to individuals with MCI as “patients”.

Response: Thank you for your valuable comment. In accordance with your advice, we have revised MCI patients to individuals with MCI consistently across our manuscript.

Comment: What is CWBSD? It needs to be briefly explained in the introduction.

Response: Thank you for your valuable comment. As per your suggestion, we have included a brief explanation of Cheonwangbosim-dan (CWBSD) in the introduction to clarify its composition and traditional applications. The revised section now states: “Cheonwangbosim-dan (CWBSD), also known as Tennohosintan in Japan and Tian Wang Bu Xin Dan in China, is a traditional herbal medicine composed of 15 medicinal herbs. It has been widely used in Korea for conditions such as insomnia, depressive and anxiety disorders, and heart palpitations. Given its potential effects on neurological health, it has been investigated for its possible role in cognitive function [19–21].” (Page 4, line 69–73)

Comment: Methods

• How was 3g dose of CWBSD administered? Was it a capsule/tablet ? divided dose? This needs to be elaborated

Response: Thank you for your query. We have elaborated on the administration method of CWBSD in the intervention subsection. The revised section now states: “The CWBSD pellet, a standardized extract, is a brown powdered mixture of medicinal herbs formulated into a solid pellet form. CWBSD (Product Name: Soon Shim Hwan) is an over-the-counter (OTC) drug, packaged in 3g per dose, and is taken orally as a single daily administration. Participants were instructed to swallow the pellet whole with water. Detailed components, appearance, and packaging are presented in the S5 file. (Page 8, line 167–171)

Comment: Physical exercise can have a profound impact on mental function. What kind of exercise was included in the study?

Response: Thank you for your query. At each visit, participants received 10–20 minutes of individualized education from a researcher using a brochure. The educational program provided fundamental information on MCI, dietary recommendations, lifestyle modifications, and exercise guidelines to promote cognitive health. This program was based on the Korean dementia guidebook published by the Ministry of Health and Welfare. The exercise component of the program emphasized establishing regular exercise habits, such as daily mild stretching and toning exercises and walking for at least 30 minutes, and instructed participants in simple exercises focusing on stretching, balance, coordination, and relaxation to help maintain cognitive function. We have elaborated on the type of exercise included in the study in the intervention subsection. The revised section now states:

“The educational program included recommendations for exercise, diet, and lifestyle management to support cognitive health. The exercise component of the program emphasized establishing regular exercise habits, such as daily mild stretching and toning exercises and walking for at least 30 minutes, and instructed participants in simple exercises focusing on stretching, balance, coordination, and relaxation to help maintain cognitive function. ” (Page 8–9, line 185–189)

Comment: It is also well known that food habits affect mental performance. The diet and lifestyle management need to be elaborated.

Response: Thank you for your valuable comments. Our education program also included diet and lifestyle management to prevent cognitive decline. Regarding diet, we recommended consuming foods that support cognitive function, such as fruits, fatty fish, whole grains, vegetables, soy products, and legumes, while limiting intake of sweeteners, unhealthy fats, and refined carbohydrates that can impair cognitive function. Lifestyle management involved managing underlying conditions that can contribute to cognitive impairment, such as hypertension, diabetes, and dyslipidemia, as well as stress management, cognitive training, abstaining from smoking and excessive alcohol consumption, and maintaining close social ties with family and others. We revised the intervention subsection accordingly as follows: “ Dietary recommendations emphasized the consumption of nutrient-rich foods beneficial for cognitive function, such as fruits, fatty fish (e.g., salmon, mackerel, sardines), whole grains, vegetables, soy products, and legumes, while advising participants to limit the intake of sweeteners, unhealthy fats, and refined carbohydrates that may negatively affect cognition. Lifestyle management focused on modifiable risk factors, including the management of underlying conditions such as hypertension, diabetes, and dyslipidemia, stress management, cognitive training, smoking cessation, alcohol avoidance, and maintaining social engagement with family and community members. These guidelines were delivered through individual educational sessions (10–20 minutes) during each visit, supplemented by a standardized brochure based on the dementia guidebook published by the Ministry of Health and Welfare in Korea. However, adherence to these recommendations was not actively monitored during the study.” (Page 9, line 189–200)

Comment: In a 24-week study, how was compliance ensured? Were the participants contacted at regular intervals to remind them to take the supplements?

Response: Thank you for your query. We ensured participant compliance through structured follow-ups and medication monitoring. The median medication compliance in both the CWBSD and control groups was 96.92%. To facilitate adherence, the CRC provided detailed explanations of visit schedules during each visit and reminded participants of upcoming appointments via phone calls one day prior to their visits. Participants visited the clinic every six weeks and received a six-week supply of the trial medication in a sealed box. To further ensure compliance, the CRC conducted at least one phone check between visits to monitor medication intake. Participants were also required to return any unused medication during their second to fifth visits, allowing adherence to be assessed through pill counts. We have revised the intervention subsection as follows: “The clinical research coordinator (CRC) conducted at least one phone check between visits to monitor medication intake, and participants were instructed to return any unused medication during their second to fifth visits. Adherence was assessed through pill counts to ensure compliance.” (Page 8, line 178–181)

Comment: How was safety ensured since the participants visited the hospital only after 12 weeks.

Response: Thank you for your comments. We ensured participant safety through regular monitoring of adverse events and physiological parameters. Although hospital visits were scheduled every 6 weeks, safety was monitored continuously throughout the study. Participants were instructed to report any adverse events (AEs) or serious adverse events (SAEs) immediately to the clinical research coordinator (CRC) via phone. Additionally, the CRC conducted periodic phone follow-ups between visits to check for any health concerns or unexpected symptoms. After 24 weeks of treatment, we assessed safety by comparing blood chemistry parameters, blood pressure, pulse rate, and the incidence of AEs/SAEs between the two groups. No SAEs or drug-related AEs were reported. No significant drug-induced changes were observed in blood pressure, pulse rate, or blood chemistry parameters, except for a significant increase in potassium levels in the CWBSD group. However, this increase remained within the normal range. Given the study's duration and funding limitations, post-treatment follow-up was not conducted. Future studies are needed to evaluate the long-term safety of CWBSD in MCI patients beyond 24 weeks. We have revised the abstract and conclusion sections as follows:

Abstract

“We demonstrated that CWBSD treatment may be safe and did not significantly improve cognitive function, quality of life, daily activities, or depression among individuals with MCI.” (Page 2, line 41–44)

Conclusion

“Our results suggest that a 24-week course of CWBSD treatment was well tolerated, with no observed drug-related adverse events or significant drug-induced alterations in blood pressure, pulse rate, or blood chemistry parameters.” (Page 20, line 411–413)

Comment: Results

At what stage did the participants drop out of the trial. Only 38 participants completed the study. The numbers used for analysis is very small. How could the data for 48 participants be used for efficacy analysis?

Response: Thank you for your query. Six participants in the CWBSD group and four in the control group withdrew from the study during the medication phase after being assigned to their respective groups due to withdrawal of consent. To address missing data, efficacy analysis was performed using a full analysis set (FAS), and missing values were imputed using the last observation carried forward (LOCF) technique. LOCF is a widely used statistical method in longitudinal studies where missing follow-up observations are replaced with the participant’s last recorded value. This approach assumes that missing data are missing completely at random (MCAR) and helps maintain statistical power despite dropouts.

We have revised the statistical analyses and participants subsections as follows: “Efficacy evaluation was performed using a full analysis set. As missing data were assumed to be missing completely at random (MCAR), missing values were imputed using the last observation carried forward (LOCF) technique.” (Page 12, line 263–266)

“Six participants in the CWBSD group and four in the control group dropped out during the medication phase because of withdrawal of consent.” (Page 13, line 283–285)

Comment: Safety data on biochemistry and hematology should be included at least as a supplementary table

Response: Thank you for your valuable suggestion. In accordance with your suggestion, we have inserted safety data on biochemistry and hematology as supplementary file S6.

Comment: In 24 weeks only 2 participants reported an adverse event? Were all events captured?

Response: Thank you for your query. All adverse events (AEs) and serious adverse events (SAEs) were carefully monitored and documented throughout the study. Participants were explicitly asked about any adverse symptoms at each visit, and all reported AEs were recorded in detail by the clinical research coordinator (CRC), including the time of occurrence, severity, potential causal relationship to the trial medication, and any treatment received. Additionally, participants were encouraged to report any unexpected symptoms between visits via phone contact with the CRC. The CRC reported all recorded AEs and SAEs to the principal investigator and the Institutional Review Board (IRB) for appropriate action. We have revised the safety outcome subsection as follows:

“All AEs and SAEs were actively monitored throughout the study. Participants were systematically asked about any adverse symptoms at each visit, and they were encouraged to report any unexpected symptoms between visits via phone contact with the CRC. Reported AEs were documented in detail, including the time of occurrence, severity, potential causal relationship to the trial medication, and any treatment provided. The CRC reported all recorded AEs and SAEs to the principal investigator and the IRB to ensure appropriate follow-up and action.” (Page 11, line 243–249)

Comment: Discussion

There are several limitations in the study. Although age was significantly higher in CWBSD, all the baseline scores of outcome parameters were almost the same. So this could not be a reason for t CWBSD group not exhibiting a significant improvement.

Response: Thank you for your valuable feedback. In response to your suggestions, we have removed the content in the discussion section that the participants’ age may have affected the results.

Comment: The exercise and diet could have had an effect in both groups. It would have made an impact to correlate the effect of these factors on cognitive improvement. May be use them as covariate for the analysis.

Response: Thank you for your feedback. Since education on daily life management and exercise was provided to both groups, but adherence was not systematically tracked, it was not possible to include these factors as covariates in the analysis. However, we acknowledge that diet and exercise could have had an impact on cognitive outcomes in both groups.

In response to your suggestion, we have revised the discussion section as follows: “First, the education on self-management and exercise provided to both groups may have influenced the results. Lifestyle modifications, including diet and physical activity, play a crucial role in managing MCI [38], and regular exercise is recommended in clinical practice guidelines for improving cognitive function in individuals with MCI [11]. However, as adherence to these lifestyle interventions was not actively monitored, it was not feasible to include them as covariates in our analysis. Future studies should consider systematically tracking diet and exercise adherence to assess their independent effects on cognitive outcomes.” (Page 19, line 374–381)

Comment: A dose of 3g/day is high. Increasing the dose can have other deleterious effects

Response: Thank you for your feedback. We fully understand your concerns regarding the potential adverse effects of increasing the CWBSD dose. The approved therapeutic dose of CWBSD in the Republic of Korea is 3g/day, and its safety profile at this dosage has been established based on traditional usage and regulatory standards. However, increasing the dose beyond this level may introduce unknown risks, as higher doses have not been systematically evaluated in clinical settings.

In response to your suggestions, we have removed the content in the discussion section that a daily dose may have i

---

## [Decision Letter · Decision Letter 3]

PONE-D-23-33277R3The efficacy and safety of Cheonwangbosim-dan (Tian Wang Bu Xin Dan) for treatment of mild cognitive impairment : a randomized placebo-controlled pilot trialPLOS ONE

Dear Dr. Kim,

Thank you for submitting your manuscript to PLOS ONE. After careful consideration, we feel that it has merit but does not fully meet PLOS ONE’s publication criteria as it currently stands. Therefore, we invite you to submit a revised version of the manuscript that addresses the points raised during the review process.

We look forward to receiving your revised manuscript.

Kind regards,

Muhammad Salman Bashir, M.S.C

Academic Editor

PLOS ONE

Reviewers' comments:

Reviewer's Responses to Questions

**Comments to the Author**

1. If the authors have adequately addressed your comments raised in a previous round of review and you feel that this manuscript is now acceptable for publication, you may indicate that here to bypass the “Comments to the Author” section, enter your conflict of interest statement in the “Confidential to Editor” section, and submit your "Accept" recommendation.

Reviewer #1: (No Response)

2. Is the manuscript technically sound, and do the data support the conclusions?

Reviewer #1: No

3. Has the statistical analysis been performed appropriately and rigorously? 

Reviewer #1: No

4. Have the authors made all data underlying the findings in their manuscript fully available?

Reviewer #1: Yes

5. Is the manuscript presented in an intelligible fashion and written in standard English?

Reviewer #1: Yes

6. Review Comments to the Author

Reviewer #1: Line 265-266: Last Observation Carried Forward (LOCF) technique is not recommended if the data is Missing Completely at Random (MCAR).

7. PLOS authors have the option to publish the peer review history of their article (what does this mean? ). If published, this will include your full peer review and any attached files.

**Do you want your identity to be public for this peer review?** For information about this choice, including consent withdrawal, please see our Privacy Policy .

Reviewer #1: No

---

## [Author Response · Author response to Decision Letter 4]

18 Apr 2025

We are very grateful for the constructive comments and the opportunity to revise our manuscript.

Point-by-point responses to the reviewers’ comments are provided below. The corresponding changes have been highlighted in the revised manuscript with track changes.

Reviewer #1

Comment: Line 265-266: Last Observation Carried Forward (LOCF) technique is not recommended if the data is Missing Completely at Random (MCAR).

Response: Thank you for your valuable comments. In clinical trials or some longitudinal studies, it is inevitable that missing values will occur. Several statistical approaches have been applied to the analysis of longitudinal data with missing values. The most commonly used imputation methods include complete case analysis (CCA), mean imputation, last observation carried forward (LOCF), hot deck, regression imputation, K-nearest neighbor, the expectation maximization algorithm, and multiple imputation. CCA assumes missing completely at random(MCAR) and includes only those with observed outcome data in the analysis. When the size of the dataset is large enough, analysis could be considered using complete case analysis method where a subject is completely deleted whenever this subject has missing values at any measurement occasion. This causes loss of potentially valuable information about the incomplete cases and MCAR may be hard to justify. Investigators can impute values for missing outcome data rather than ignoring them. The simplest imputation method is the LOCF method that substitutes every missing value with its corresponding last observed value. The LOCF method is a very common approach for handling missing data especially in dropout missingness under MCAR mechanism. This method assumes that the result would not change after the last observed value. Thus, there is no time effect since the last observed value. This situation could be considered as unrealistic in many settings. Thus, LOCF method tends to underestimate the true variability of the data. Compared to CCA method, LOCF method maintains the sample size. However, LOCF method may include bias when dealing with a longitudinal dataset in addition to the long time point of measurement in each interval [1,2].

These approaches should be selected based on the amount of missingness and the missingness mechanism. Some statistical methods are valid only under certain situations with specified missing rates. In other words, there is no unique best method available for all situations.

Despite concerns about the relatively low power and high bias of the LOCF method, we set up a statistical analysis method in the protocol by referring to the statistical analysis method of the previous clinical trial of herbal medicine for mild cognitive impairment [3,4].

As per your suggestion, we have revised the statistical analyses subsection as follows: “Statistical analysis protocol was developed by referring to previous clinical trials of herbal medicine for MCI [38, 39]. Statistical analyses were revised according to IRB directives during the trial period. Efficacy evaluation was performed using a full analysis set. Missing values were imputed using the last observation carried forward (LOCF) technique.” (Page 12, line 263-266)

Reference

1. GAD, Ahmed Mahmoud; ABDELKHALEK, Rania Hassan Mohamed. Imputation methods for longitudinal data: A comparative study. International Journal of Statistical Distributions and Applications. 2017; 3(4): 72.

2. Nakai M, Chen DG, Nishimura K, Miyamoto Y. Comparative study of four methods in missing value imputations under missing completely at random mechanism. Open Journal of Statistics. 2014; 4(1): 27-37.

3. Tian J, Shi J, Li T, Li L, Wang Z, Li X, Lv Z, Zheng Q, Wei M, Wang Y. Efficacy and Safety of an Herbal Therapy in Patients with Amnestic Mild Cognitive Impairment: A 24-Week Randomized Phase III Trial. Evid Based Complement Alternat Med. 2017;2017:4251747. doi: 10.1155/2017/4251747.

4. Shin HY, Yim TB, Heo HM, Jahng GH, Kwon S, Cho SY, Park SU, Jung WS, Moon SK, Ko CN, Park JM. Effects of Kami Guibi-tang in patients with mild cognitive impairment: study protocol for a phase III, randomized, double-blind, and placebo-controlled trial. BMC Complement Med Ther. 2022 Dec 2;22(1):318. doi: 10.1186/s12906-022-03805-9.

---

## [Decision Letter · Decision Letter 4]

The efficacy and safety of Cheonwangbosim-dan (Tian Wang Bu Xin Dan) for treatment of mild cognitive impairment : a randomized placebo-controlled pilot trial

PONE-D-23-33277R4

Dear Dr. Kim,

We’re pleased to inform you that your manuscript has been judged scientifically suitable for publication and will be formally accepted for publication once it meets all outstanding technical requirements.

Kind regards,

Muhammad Salman Bashir, M.S.C

Academic Editor

PLOS ONE

Additional Editor Comments (optional):

Reviewers' comments:

Reviewer's Responses to Questions

**Comments to the Author**

1. If the authors have adequately addressed your comments raised in a previous round of review and you feel that this manuscript is now acceptable for publication, you may indicate that here to bypass the “Comments to the Author” section, enter your conflict of interest statement in the “Confidential to Editor” section, and submit your "Accept" recommendation.

Reviewer #1: All comments have been addressed

Reviewer #4: All comments have been addressed

2. Is the manuscript technically sound, and do the data support the conclusions?

Reviewer #1: (No Response)

Reviewer #4: (No Response)

3. Has the statistical analysis been performed appropriately and rigorously? 

Reviewer #1: (No Response)

Reviewer #4: (No Response)

4. Have the authors made all data underlying the findings in their manuscript fully available?

Reviewer #1: (No Response)

Reviewer #4: (No Response)

5. Is the manuscript presented in an intelligible fashion and written in standard English?

Reviewer #1: (No Response)

Reviewer #4: (No Response)

6. Review Comments to the Author

Reviewer #1: Line 263-266: The statement could be improved. e.g.

"The statistical analysis protocol for this study was developed with the reference to the previous clinical trials on herbal medicine for mild cognitive impairment (MCI), which also utilized the LOCF technique for handling missing data [38, 39]. During the trial period, statistical analyses were revised in accordance to the IRB directives. Efficacy was evaluated using the full analysis set, and the missing data were imputed using the LOCF technique to maintain the consistency with the methodology and also to ensure that it is comparable with the prior findings"

Reviewer #4: (No Response)

7. PLOS authors have the option to publish the peer review history of their article (what does this mean? ). If published, this will include your full peer review and any attached files.

**Do you want your identity to be public for this peer review?** For information about this choice, including consent withdrawal, please see our Privacy Policy .

Reviewer #1: No

Reviewer #4: No

---

## [Editor Report · Acceptance letter]

PONE-D-23-33277R4

PLOS ONE

Dear Dr. Kim,

I'm pleased to inform you that your manuscript has been deemed suitable for publication in PLOS ONE. Congratulations! Your manuscript is now being handed over to our production team.

Kind regards,

on behalf of

Dr. Muhammad Salman Bashir

Academic Editor

PLOS ONE